# Collaborative Learning under Strategic Behavior: Mechanisms for Eliciting Feedback in Principal-Agent Bandit Games

## Abstract

Collaborative multi-armed bandits (MAB) has emerged as a promising framework that allows multiple agents to share the *burden of exploration* and find optimal actions for a common problem. While several recent results demonstrate the benefit of collaboration in minimizing per-agent regret, prior work on collaborative MABs primarily rely on the assumption that all participating agents behave truthfully. The case of *strategic* agent behavior where an agent may *free-ride* on the information shared by others without performing exploration has received limited attention in the collaborative MAB setting; such free-riding strategies can lead to a collapse in exploration resulting in high regret for all honest agents.

This paper addresses the problem of collaborative multi-armed bandits in the presence of strategic agent behavior. Our main contribution is to design mechanisms for penalizing agents so that truthful behavior, i.e., performing sufficient exploration and reporting feedback accurately, is a Nash equilibrium. Furthermore, under this Nash equilibrium, the per-agent regret with collaboration is $\sqrt{M}$-factor smaller than the per-agent regret without collaboration, where $M$ is the number of agents. Our results establish that it is possible to achieve the benefit of collaboration even in the presence of strategic agents who may want to free-ride. Semi-synthetic experiments show that our theoretical results hold empirically, as well.

## 1. Introduction

Multi-armed bandits is a framework to study problems where an agent is repeatedly faced with the *explore-exploit* dilemma, i.e., to acquire new information or to select optimal actions based on existing information (Lattimore & Szepesvári, 2020). This framework is used in a wide range of applications including clinical trials (Thompson, 1933; Villar et al., 2015), recommendation systems (Li et al., 2010), advertising (Buccapatnam et al., 2013), dynamic pricing in markets (Trovò et al., 2018), portfolio selection (Huo & Fu, 2017), hedging (Cannelli et al., 2023), sponsored search auctions (Sharma et al., 2012) and resource allocation (Krishnasamy et al., 2021). In recent years there has been increasing interest in *collaborative* multi-armed bandits (Dubey & Pentland, 2020; Wang et al., 2019; Li & Wang, 2022; He et al., 2022). This interest is fuelled by the promise of data sharing and interoperability leading to better individual and societal utility (Mancini, 2021). In this setting, multiple agents can collaborate on a common bandit task by sharing data across a communication channel. This allows the agents to divide the *burden of exploration* amongst themselves, and find optimal actions based on the collective information. For example, multiple hospitals can perform joint exploration in the clinical trial of a new vaccine (Rieke et al., 2020; Flores et al., 2021), several NGOs and government agencies can collaborate with each other to perform better mobile interventions for healthcare (Ou et al., 2022), multiple city administrations can collaborate to solve common challenges such as health inspections and waste management (Mao & Perrault, 2024), autonomous driving providers can benefit from joint exploration of routes and safety conditions (Ferdowsi et al., 2019), and finally, several recommendation platforms can share aggregate information to improve exploration and better optimize their content using the bandit framework (Bouneffouf et al., 2020). Recent work has shown that this ability to collaborate leads to a significant reduction in per-agent regret in various settings (Wang et al., 2019).

However, almost all these results rely on the assumption that all agents duly contribute in exploration and share their feedback truthfully. This assumption is presumably idealistic and can be easily violated in practice due to the presence of self-interested *strategic* agents which might be tempted to 'free-ride' on the information shared by others. In particular, these agents might purely rely on exploitation of actions that are explored by other agents. In fact, (Jung et al., 2020) showed that if an agent is able to observe the actions and rewards of another regret-minimizing agent, then there exists a 'free-riding' strategy which suffers negligible re-

gret. Hence, the existence of such strategies can lead to a collapse in exploration as many agents might be tempted to adopt this strategy. Moreover, agents might be unwilling to report their accurate reward estimates without the presence of any incentives. Despite its practical relevance, the presence of "free-riding" strategic agents has been relatively unexplored in the collaborative multi-armed bandits literature. In this work, we ask therefore the following:

> **Research Question.** *Is it possible to incentivize strategic agents to participate truthfully, and not "free-ride" in a collaborative multi-armed bandits setup?*

To understand this question, we combine two areas of research: mechanism design and multi-agent multi-armed bandits. In particular, we consider mechanisms for a principal to incentivize truthful participation in a linear stochastic multi-armed bandits setup with $M$ agents. Each agent is faced with the same linear bandit instance with time-horizon $T$ and the goal of each agent is to minimize their own regret (Abbasi-Yadkori et al., 2011). We use monetary penalty as the main device by which the principal can incentivize truthful behavior. The principal acts as a central coordinator who shares/collects information to/from agents and also enforces the mechanism to penalize the agents. The goal of the principal is to minimize the social regret, i.e., the cumulative regret of all agents. The communication between the agents and principal happen in synchronous rounds of communication.

### 1.1. Our contributions

**Mechanism.** We design provable mechanisms, LsHon (**L**ow-**S**witching **Hon**est protocol) for the agent and MonPen (**Mon**etary **Pen**alty protocol) for the priincpal, such that truthful behavior, i.e., duly exploring the required actions and reporting the correct information, is a Nash equilibrium (Theorem 1).

**Theorem 1** (Informal, see Thm. 3)**.** *When Principal follows* MonPen *in common knowledge, all agents following* LsHon *is a Nash Equilibrium.*

To our knowledge, this is the first mechanism to disincentivize 'free-riding' in collaborative multi-armed bandits. LsHon prescribes truthful behaviour to the agents. Thus, this result shows that our protocols are Bayesian-Nash Incentive Compatible. Furthermore, we show that under this Nash equilibrium, each agent achieves an orderwise-optimal regret (Theorem 2).

**Theorem 2** (Informal, see Thm. 4)**.** *When all agents following* LsHon*, every agent enjoys order-wise optimal regret bound of $\tilde{O}(d\sqrt{T/M})$ and arbitrarily small expected penalty.*

Here, $d$ is the ambient dimension for actions. In contrast to the $\tilde{O}(d\sqrt{T})$ per-agent regret achievable without any collabo-

ration (Abbasi-Yadkori et al., 2011), our results imply a $\sqrt{M}$ reduction in per-agent regret with collaboration, even in the presence of strategic agents. This shows that our protocol mechanism is Individually Rational, i.e., the agents are better off by participating in this collaboration, despite the threat of a monetary penalty, than abstaining.

Moreover, the optimality of the per-agent regret bound implies that this Nash equilibrium is order-wise socially optimal in terms of regret, i.e., there are no hand-crafted strategy profiles for the set of agents (even among those that do not constitute an equilibrium) that can lead to a better order-wise social regret.

**Methods.** Our results are built upon several novel techniques: Firstly, MonPen performs carefully chosen statistical manipulations and employs a penalty function designed such that the expected penalty converges for an honest agent and diverges to infinity for a strategic agent. Secondly, we have LsHon make use of a low-switching bandit policy (Abbasi-Yadkori et al., 2011) for each agent that allows us to synchronize arm play so that per-agent regret of $\tilde{O}(d\sqrt{T/M})$ is possible.

**Experiments.** We complement our theoretical results with empirical results in a semi-synthetic environment which show that while "free-riding" agents can enjoy a smaller regret than truthful agents, the penalty levied by our mechanism decreases the overall utility of this behavior, to a value lower than the utility of an honest agent.

## 2. Setting & Preliminaries

We describe our setting and setup notation in this section.

**Communication setup.** We consider $M$ agents and a single principal. The communication network between these agents and principal is setup according to a star topology. More specifically, every agents can communicate with the principal by sending and receiving packets, but the agents cannot communicate amongst themselves. The communication happens synchronous over rounds, i.e., the principal will coordinate upon a time at which all agents will send packets to the principal and the principal will respond to these packets.

**Multi-agent linear bandit setting.** Each agent $a \in \mathcal{A}$ is faced with the same linear bandit instance with time-horizon $T$. At each trial/time-step $t \in [T]$, the agent needs to choose an action $x_{a,t}$ from the decision set $\mathcal{X}_t \subseteq \mathbb{R}^d$. Note that $\mathcal{X}_t$ is identical across all agents, but can vary over time. Upon playing action $x_{g,t}$ the agent observes then its reward

$$y_{a,t} = \langle \theta^*, x_{a,t} \rangle + \eta_{a,t}$$

where $\theta^* \in \mathbb{R}^d$ is an unknown parameter and $\eta_{a,t} \sim \mathcal{N}(0,1)$ is zero-mean random noise. Given the knowledge of $\theta^*$, the optimal action at trial $t$ is $x_t^* =$

$\arg\max_{x \in \mathcal{X}_t} \langle \theta^*, x \rangle$ and the expected cumulative reward of this action is given by $\sum_{t=1}^{T} \langle \theta^*, x_t^* \rangle$. We will measure the regret of the agent against this optimal strategy, i.e.,[1]

$$R_a(T) = \sum_{t=1}^{T} \langle \theta^*, x_t^* \rangle - \langle \theta^*, x_{a,t} \rangle .$$

We make the standard assumption that $\|\theta^*\|_\infty \leq 1$ and $\|x\|_2 \leq 1$ for all $x \in \mathcal{X}_t$ for all $t$.

**Strategic Agent.** A rational agent shall be tempted to not truthfully follow the given protocol LSHON and can deviate in two aspects : first, he can deliberately misreport his parameter estimate $\theta_{a,t}$, second, he can ignore the arm play algorithm and instead play actions of his choice. Specifically, the agent is said to *free-ride* if he explores 'lesser' than LSHON requires him to. (See Defn. 1)

**Agent-principal contract.** The goal of the principal is to minimize the social regret, i.e., the cumulative regret $\sum_{a \in \mathcal{A}} R_a(T)$ across all agents. However, to guard against such free-riding strategies and ensure agents act truthfully, we consider a setting where there is a contract between the agents and the principal. Under this contract the principal can levy a monetary penalty on the agent at every communication round based on their assessment of the agent's honesty. In this case, the goal of every agent $a$ is to maximize his utility, i.e., minimize the sum of his regret and the penalty levied on him:

$$U_a(T) = -R_a(T) - P_a(T),$$

where $P_a(T)$ is the overall penalty levied on agent $a$.

## 3. Algorithm and Protocol Sketch

We defer the thorough descriptions and pseudocodes of our protocols/algorithms to the Appendix D in the interest of space, and present rough sketches of their salient aspects briefly in this section.

**Agent protocol.** Consider any agent $a$. The agent protocol LSHON (**L**ow-**S**witching **Hon**est protocol) prescribes the agent to report truthfully it's parameter estimate $\theta_{a,\tau}$ at all communication events $\tau$. Further, it prescribes the agent to follow the low-switching arm play algorithm of (Abbasi-Yadkori et al., 2011) which is similar to the LinUCB algorithm except that it changes/updates its parameter estimate very infrequently. At any time $t$, the agent shall maintain a confidence set $C_{a,\tau}$ (as in eqn. 3) around the parameter estimate $\theta_{a,\tau}$ (as in eqn. 2) from all statistics from both self-play and collaboration as of the previous communication event at time $\tau$. This is said to be 'low-switching' as

the parameter estimate and confidence sets are not updated at every time-step, but only during communication events. Then, this algorithm balances exploration and exploitation by selecting an *optimistic* action based on this confidence set:

$$x_{a,t} = \arg\max_{(x,\theta') \in \mathcal{X}_t \times C_{a,\tau}} \langle x, \theta' \rangle . \quad (1)$$

This low-switching aspect ensures that the trajectory of arm plays is deterministic without being affected by reward noises between any two communication events, and also ensures that all agents perform similar amounts of exploration. These are crucial to show the $\sqrt{M}$ benefit in regret reduction. We refer the reader to Appendix E for more details.

**Principal protocol.** As per MONPEN (**Mon**etary **Pen**alty protocol), the principal shall initiate communication events at appropriate times (line 4). During a communication event, the principal receives the parameter estimates $\theta_{a,\tau}^\varepsilon$s from all agents $a$, that is possibly corrupted. The principal does two things: first, he facilitates complete sharing of information. To every agent, he sends the aggregate statistics from all other agents without withholding any. This shall ensure all agents get optimal regret. Second, he shall impose a penalty such that when all other agents are honest, agent $a$ suffers $O(1)$ penalty if he is also honest, but the penalty tends to $+\infty$ if he free-rides. This primarily hinges on the principal's ability to distinguish and honest agent and free-rider, probabilistically, without knowing the true parameter value $\theta^*$. Towards this, the principal computes agent-wise intermediate quantities, $z_a$s, as a function of all agents' estimates (see Appn D.2 for full details), that is shown to follow a standard normal distribution if $a$ is honest, but some component of $z_a$ has a marginal variance strictly more than 1 if $a$ free-rides. Taking advantage of this distinction, the principal computes the penalty of agent $a$ as function of these components $x$ as follows:

$$P = \frac{\sqrt{2\pi x}}{(x+1)^{1+c_2}} \cdot \exp\left(\frac{x}{2}\right) - \frac{1}{c_2},$$

which is specifically hand-crafted such that it's expected value converges to zero for variance 1 (for an honest agent) and doesn't converge and tends to $+\infty$ for variance more than 1 (for a dishonest agent).

## 4. Experiments

We demonstrate the effectiveness of our approach in dis-incentivizing 'free-riding' under two different linear bandit environments: (1) synthetically generated (deferred to Appendix H), (2) based on the Movielens dataset (Harper & Konstan, 2015).

---

[1] This notion of regret is sometimes referred to as *pseudo-regret* in the literature.

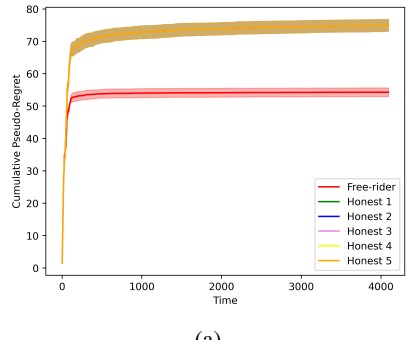
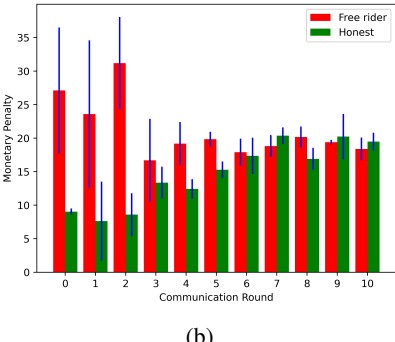
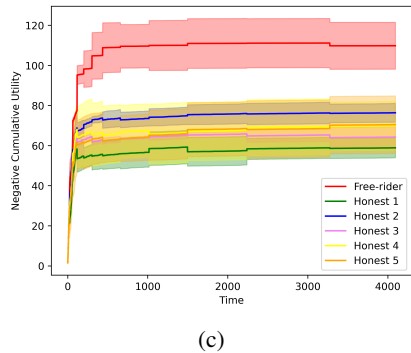

(a)                                    (b)                                    (c)

Figure 1: **MovieLens problem instance** (a) Regret of agents as a function of time $t$. (b) Penalty of agents as a function of communication rounds. (c) Negative utility of agents.

**Bandit environment setup.** The Movielens environment is motivated by a real-world movie recommendation setting where multiple recommendation platforms can share the burden of exploration while being robust to the presence of strategic agents. The action sets $\{\mathcal{X}_t\}$ are generated by jointly embedding users and movies in a $d = 100$ dimensional space such that each $\mathcal{X}_t$ corresponds to the set of movies that the platform can recommend to a given user with $|\mathcal{X}_t| = 1682$. The true $\theta^* \in \mathbb{R}^{100}$ for our linear bandits setup is the minimizer of the least square error between the predicted ratings and the true ratings.

We have an agent, say $a$, who is a free-rider. Specifically, he plays $x_{a,t} = \arg\max_{x \in \mathcal{X}_t} \langle x, \theta_{a,\tau} \rangle$ completely exploiting on the current knowledge of parameter $\theta_{a,\tau}$, while all other agents fulfill the exploration duty (as in eqn. 1).

In the experiment, the reward for playing an arm $x \in \mathcal{X}_t$ is $\langle x, \theta^* \rangle + \eta$ where $\eta \sim \mathcal{N}(0, 0.1)$. We set the number of agents to $M = 10$. We simulate our Protocol for a time horizon of $T = 4096$ for 3 repetitions. We use the linear bandits implementation provided in the Pearl library (Zhu et al., 2023) and extend it to our multi-agent setting. We set $\lambda = 1$, $c_2 = 0.1$ and $c_1 = 1.4^d$. The result of the Movielens experiment is plotted in Fig. 1.

**Experiment outcome.** Comparing the regret of the free-rider and a number of honest agents, we observe that the regret of the free-rider is lower than that of the honest agents. The exploration information that the free-rider obtains during the communication events more than sufficiently compensates for his own lack of exploration, thereby letting him enjoy a better regret. At this juncture, we note that the algorithms in the literature (whose goals were different to that of handling strategic agents) such as (Wang et al., 2019) are not immune this phenomenon of agents free-riding.

However, in our Protocol, we observe that the penalty levied by our algorithm is significantly higher for the free-rider. The principal's protocol 3 correctly penalizes the free-rider more than it does the honest agents. Hence, this combination of regret and penalty (which we refer to as the negative utility) is higher for the free-rider than for the honest agent, which makes free-riding undesirable.

It is also interesting to note the variation of penalty across the different communication events. The free-rider incurs a lot more penalty than the honest agent in the initial few rounds, whereas their penalties appear to become similar towards the later stages. This can be attributed to the inherent nature of the arm play algorithms employed by the two. The free-rider always exploits. The honest agent balances his exploration and exploitation, inherently leaning towards exploring more in the early stages and towards exploiting more in the later part. Thus, while the exploitative arm play of the free-rider starkly differs from the highly exploratory arm play of the honest agent in the initial stages resulting in heavier penalties for the free-rider, their arm plays get 'closer' to that of each other towards the later stages when both predominantly (and rightly) exploit, resulting in similar penalties.

## 5. Conclusion

This paper considers the problem of collaborative multi-armed bandits under strategic behavior and describes new multi-arm bandit protocols for the principal that induce the Nash equilibrium of desirable truthful agent behaviour. We show that one can obtain nearly the full benefit of altruistic collaboration despite strategic behaviour from agents by establishing minimax optimal regret guarantees at this equilibrium. As future work, we aim to expand our work to consider an asymmetric set of agents with different action sets and alternative communication frameworks to study the dichotomy between the benefits of collaboration and ill-effects of strategic agent behaviour.

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

# A. Preliminaries

Table 1: Description of notations frequently used in the document.

| Notation | Description |
|---|---|
| $\mathcal{A}$ | The set of all $M$ agents. |
| $V_{a,t}^{\varepsilon}$ | The actual design matrix of agent $a$ at time $t$ including samples from both self-play and collaboration. |
| $V_{a,t}^{self}$ | The actual design matrix of agent $a$ at time $t$ including samples from only self-play. |
| $V_{a,t}$ | The hypothetical design matrix (from self-play and collaboration) agent $a$ will have at time $t$ when all agents adhere to LsHon. |
| $\theta_{a,t}$ | Agent $a$'s actual estimate of $\theta^*$ from both collaboration and self-play, at time $t$. |
| $\tau$ | Arbitrary time-step of communication. |
| $\theta_{a,t}^{\varepsilon}$ | The estimate that agent $g$ reports to the Principal in a communication event at time $t$. |

## A.1. Math Fundamentals

**Claim 1** (Probability density of Generalized $\chi^2(1)$)**.** *Let $X \sim \mathcal{N}\left(\mu, \sigma^2\right)$ be a univariate gaussian random variable. The probability density function of $Y = X^2$ is given by*

$$f_Y(y) = \frac{1}{\sigma\sqrt{2\pi y}} \exp\left(\frac{-1}{2} \cdot \frac{y + \mu^2}{\sigma^2}\right) \times \left[\exp\left(\frac{\sqrt{y}\mu}{\sigma^2}\right) + \exp\left(\frac{-\sqrt{y}\mu}{\sigma^2}\right)\right],$$

*when $y \geq 0$, and $0$ otherwise.*

*Proof.* The PDF of $X$ is given by $f_X(x) = \frac{1}{\sigma\sqrt{2\pi}} \exp\left(\frac{-1}{2}\left(\frac{x-\mu}{\sigma}\right)^2\right)$. The CDF of $Y$ can be written as

$$F_Y(y) = \mathbb{P}\left\{Y \leq y\right\} = \mathbb{P}\left\{-\sqrt{y} \leq X \leq \sqrt{y}\right\} = F_X(\sqrt{y}) - F_X(-\sqrt{y}),$$

for any $y \geq 0$, and $0$ otherwise (when $y < 0$). Then, the PDF of $Y$ can be derived as follows:

$$\begin{aligned}
f_Y(y) &= \frac{d}{dy}\left[F_X(\sqrt{y}) - F_X(-\sqrt{y})\right] \\
&= \frac{1}{2\sqrt{y}}\left[f_X(\sqrt{y}) + f_X(-\sqrt{y})\right] \\
&= \frac{1}{2\sqrt{y}}\left[\frac{1}{\sigma\sqrt{2\pi}} \exp\left(\frac{-1}{2}\left(\frac{\sqrt{y}-\mu}{\sigma}\right)^2\right) + \frac{1}{\sigma\sqrt{2\pi}} \exp\left(\frac{-1}{2}\left(\frac{-\sqrt{y}-\mu}{\sigma}\right)^2\right)\right] \\
&= \frac{1}{2\sqrt{y}}\left[\frac{1}{\sigma\sqrt{2\pi}} \exp\left(\frac{-1}{2}\left(\frac{\sqrt{y}-\mu}{\sigma}\right)^2\right) + \frac{1}{\sigma\sqrt{2\pi}} \exp\left(\frac{-1}{2}\left(\frac{-\sqrt{y}-\mu}{\sigma}\right)^2\right)\right] \\
&= \frac{1}{2\sigma\sqrt{2\pi y}}\left[\exp\left(\frac{-1}{2\sigma^2}\left(y + \mu^2 - 2\sqrt{y}\mu\right)\right) + \exp\left(\frac{-1}{2\sigma^2}\left(y + \mu^2 + 2\sqrt{y}\mu\right)\right)\right] \\
&= \frac{1}{2\sigma\sqrt{2\pi y}} \exp\left(\frac{-1}{2\sigma^2}\left(y + \mu^2\right)\right)\left[\exp\left(\frac{\sqrt{y}\mu}{\sigma^2}\right) + \exp\left(\frac{-\sqrt{y}\mu}{\sigma^2}\right)\right].
\end{aligned}$$

This completes the proof of the Claim. Additionally, the final expression can be concisely written using the hyperbolic cosine function as follows.

$$f_Y(y) = \frac{1}{\sigma\sqrt{2\pi y}} \exp\left(\frac{-1}{2\sigma^2}\left(y + \mu^2\right)\right) \cdot \cosh\left(\frac{\sqrt{y}\mu}{\sigma^2}\right).$$

$\square$

The corollary follows.

**Corollary 1** (Probability density of $\chi^2(1)$). *When $X \sim \mathcal{N}(0, 1)$ is a standard normal random variable, $Y = X^2$ follows a chi-squared distribution of $1$ degree of freedom whose probability density is given by*

$$f_Y(y) = \frac{1}{\sqrt{2\pi y}} \exp\left(\frac{-y}{2}\right),$$

*when $y \geq 0$, and $0$ otherwise.*

## B. Technical Lemmas

In this section, we restate some lemmas from the literature that we shall use.

**Lemma 2** (Lemma 11 of (Abbasi-Yadkori et al., 2011)). *Write $V_t := \gamma I + \sum_{s=1}^{t-1} x_s x_s^\top$ for all $t \in [T]$, where $\|x_s\|_2 \leq 1$ for all $s \in [T]$. Then,*

$$\sum_{t=1}^{T} \min\left(1, \|x_t\|_{V_t^{-1}}^2\right) \leq 2 \log\left(\frac{\det(V_{T+1})}{\det(V_0)}\right) \leq 2d \log\left(\frac{d\gamma + T}{d\gamma}\right) = \tilde{O}\left(d \log T\right).$$

**Lemma 3** (Lemma 12 of (Abbasi-Yadkori et al., 2011)). *Let $A, B,$ and $C$ be positive semi-definite matrices such that $A = B + C$. Then,*

$$\sup_{x \neq 0} \frac{x^\top A x}{x^\top B x} \leq \frac{\det(A)}{\det(B)}.$$

## C. Related Work

**Stochastic linear bandits.** The stochastic multi-armed bandits problem has a rich history of development dating back to the work of (Thompson, 1933). The seminal result of (Auer et al., 2002) shows that the UCB algorithm which is based on the principle of 'optimism in the face of uncertainty' achieves a $\tilde{O}(\sqrt{KT})$ regret bound for $K$-armed bandits. The linear bandits problem has also received significant attention due to applications where each arm/action has side information which can be utilized in reward estimation (Wang et al., 2021; Moradipari et al., 2021). In the stochastic linear bandits problem, each action is embedded in a $d$ dimensional space and the rewards are a linear (stochastic) function of these action embeddings. The principle of optimism has been extended to this setting to obtain a $\tilde{O}(\sqrt{dT \ln K})$ regret bound (Chu et al., 2011), and a $\tilde{\Theta}(d\sqrt{T})$ regret bound (Dani et al., 2008). The latter bound is independent of the number of actions $K$ and allows for an infinite number of arms. Finally, (Abbasi-Yadkori et al., 2011) proposed the OFUL algorithm (also loosely referred to as LinUCB) which is based on the same principle of optimism and achieves a bound of $\tilde{O}(d\sqrt{T})$ even when the (possibly infinite) set of actions changes over time.

**Multi-agent settings.** Multi-agent multi-armed bandits have similarly been studied under different settings. Motivated by applications in wireless communication (Kalathil et al., 2014) study a multi-agent bandits with 'collision' problem. In this,if several agents play the same arm at any given time (collide), then these agents receive no reward (or sometimes a shared reward). The agents are oblivious to each other's presence with no explicit/reliable communication channel, and perceive the presence of other agents only through such collisions from their observe rewards.

Many other works have considered variations of this basic setup, for example, allowing agents to enter and exit at different times (Rosenski et al., 2016), and allowing arms to be heterogeneous (with different reward means to different agents) (Bistritz & Leshem, 2018), explicitly informing agents of collision occurrence (Bubeck et al., 2020), and designed algorithms with regret guarantees.

Another rich line of work assumes agents are co-operative and communicate via an orchestrating Principal to share helpful information towards achieving a common goal such as regret minimization, best-arm identification etc. (Hillel et al., 2013) introduced collaborative learning in stochastic bandits in the the best-arm-identification (or pure exploration) problem. With $M$ agents who can get together only once to communicate (i.e., $R = 2$ rounds[2] of play), a $\sqrt{M}$ speed-up (in terms

---

[2]Typically, the word time-step and round are used interchangeably to denote one arm play and observation. However, here, we reserve *round* to mean the series of time-steps between two communication events.

of sample complexity) per agent is achieved. (Tao et al., 2019) formalize this setting further and generalize this result to achieve a speed-up of $M^{R/R-1}$ with $R$ rounds of play. (Wang et al., 2019) comprehensively cover the regret minimization problem in multi-agent (or distributed) linear bandits with communication guarantees. They show an arm elimination-based protocol for a fixed action set that achieves $\tilde{O}(d\sqrt{TM})$ regret (overall across all $M$ agents). Further, similar to our setting, they also look at time-varying action sets and propose a Distributed Linear UCB (DisLinUCB) algorithm that extends the OFUL algorithm of (Abbasi-Yadkori et al., 2011) and achieves a regret bound of $\tilde{O}(d\sqrt{MT})$. DisLinUCB is shown to have a communication cost of $O(M^{1.5}d^3)$ that is independent of $T$ by using a specific dynamic condition to trigger communication events.

In addition, there have been several variants and extensions of the communication model, for example, peer-peer messaging (instead of a full broad-cast) (Korda et al., 2016; Szorenyi et al., 2013), restricted message sizes (Sankararaman et al., 2019) which prohibits sharing information about *all* arms, and asynchronous communication (Li & Wang, 2022; He et al., 2022) where each agent independently and individually communicates with the Principal at different times based on their local information, as opposed to having a dedicated communication event for all agents.

Furthermore, several works have considered environments that are heterogeneous to different agents, i.e., where each agent, say $a$, faces a different bandit instance, like, different linear models $\theta_a^*$ (Moradipari et al., 2022; Ghosh et al., 2022) or sets of independent arms (Shi et al., 2021; Karpov & Zhang, 2022).

Finally, there has also been work on the collaborative multi-armed bandits in the presence of adversarial agents who actively try to disrupt the collaboration (Awerbuch & Kleinberg, 2008; Vial et al., 2021; Mitra et al., 2022). The work of (Mitra et al., 2022) considers a linear bandit setup similar to ours. However, there are differences in motivation/techniques between our work and (Mitra et al., 2022): they design an algorithm that is robust to adversarial agents by computing robust reward statistics, while we consider the design of mechanisms for incentivizing strategic agents to perform sufficient exploration and report truthfully.

**Strategic agents.** A more recent theme in multi-armed bandits is to study strategic behavior of agents in a collaborative environment. In collaborative learning setups, the presence of such strategic agents brings about a notion of 'free-riding' where an agent gains from the effort of others and doesn't contribute back to the collaboration. Specifically, if an agent $x$ is permitted to observe the actions, contexts, and rewards of other agents (all of whom are assumed to be running no-regret algorithms themselves) at every time-step, (Jung et al., 2020) show that $x$ can achieve $O(1)$ regret (w.r.t. $T$, but with an $1/\Delta$ factor otherwise) by estimating its own reward means from observations from other agents alone. They further argue that the knowledge of other agents' contexts are necessary for constant order regret by showing an $\Omega(\log T)$ lower bound in that case.

In the multi-agent bandits with 'collision' problem, if several agents play the same arm (collide), the reward from that arm is 'shared' among those agents. (Xu et al., 2023; Boursier & Perchet, 2020) show existence of $\varepsilon$-Nash Equilibrium behaviours with regret guarantees in such settings. (Wei et al., 2023) suppose that every agent incurs some private cost to share information with the Principal and compares it against it's utility from getting information from other agents and consequently decides if it should participate in data sharing or not. They show that there exists private cost sequences which make any data sharing mechanism of the Principal not individually rational to participate in. To overcome this, (Wei et al., 2023; 2024) propose additional monetary payouts (that add to an agent's utility) based on honestly reported private costs of the agents to make all agents participate and achieve optimal $O\left(d/\sqrt{TM}\right)$ per-agent per-time-step regret rate.

Beyond multi-armed bandits, a closely related line of work (Karimireddy et al., 2022; Murhekar et al., 2024) considers mechanisms to incentivize data sharing in a supervised learning setting. However, these works consider the cost of acquiring and sharing data and designs mechanisms with payments to incentivize agents towards desired behaviour of acquiring and sharing data.

## D. Complete Description of Our Mechanism and Algorithm

In this section, we introduce protocol LSHON for the agent behaviour (Algorithm 2) and the protocol MONPEN for principal behaviour (Algorithm 3) that fit into the specifications of our setting Framework 1, and give a complete description of them.

---

**Algorithm 1** A multi-agent linear bandits framework with rational agents

---

1: **for** time-step $t = 1, 2, \ldots, T$ **do**
2:     **for all** agents $a \in \mathcal{A}$ **do**
3:         [Agent] Chooses and plays arm $x_{a,t}$ and observes reward $y_{a,t}$.
4:         **if** Communication event occurs **then**
5:             $\tau \leftarrow t$.
6:             [Agent] Sends current estimate $\theta_{a,\tau}$ to principal.
7:             [Principal] Sends aggregated statistics $V_{a,\tau}^P, B_{a,\tau}^P$ to agent $a$.
8:             [Principal] Levies penalty $P_{a,\tau}$ on agent $a$.
9:         **end if**
10:     **end for**
11: **end for**

---

**Framework illustration.** We illustrate in Framework 1 the interaction between various components of our system and the data transferred. Here, all agents individually play the bandit instance with their locally then available statistics (line 3). The actual arms played and feedback/reward observed are the private information of that agent and is not automatically known to other agents or the principal. At certain time-steps, say $\tau$, the principal initiates communication events (line 4). During a communication event, the agents are to send their current parameter estimates $\theta_{a,\tau}$s (line 6) to the principal, and then, the principal sends back aggregated statistics (design and results matrices) back to the agents and also levies a monetary penalty $P_{a,\tau}$ (lines 7-8). Note that each agent receives identical information but different penalties from the principal. Further, note that this framework is similar to (Wang et al., 2019), with some differences in the way a communication event is initiated.

### D.1. Agent Protocol

As part of the contract, the agent is required to adhere to LsHON, which prescribes to the agent to report his current parameter estimate $\theta_{a,\tau}$ truthfully to the principal (line 6), and prescribes to choose and play actions in the following way. The choice of actions is based on the low-switching algorithm of (Abbasi-Yadkori et al., 2011) which is similar to the LinUCB algorithm except that it changes/updates its parameter estimate very infrequently. This algorithm balances exploration and exploitation by constructing a confidence set around the current parameter estimate and selecting an *optimistic* action based on this confidence set.

---

**Algorithm 2** LsHON - Agent's truthful Protocol with low-switching arm play.

---

**Output:** The arm $x_{a,t}$ to play at time $t$. The parameter estimate $\theta_{a,\tau}^\varepsilon$ to report during communication events.
1: Most recent communication time $\tau = 0$.
2: **for** time-step $t = 1, 2, \ldots, T$ **do**
3:     Compute $\theta_{a,\tau}$ and $C_{a,\tau}$ using Eqn. 2 and 3 based on statistics as of last communication event.
4:     [Agent] Choose to play arm
$$x_{a,t} = \arg\max_{(x,\theta') \in \mathcal{X}_t \times C_{a,\tau}} \langle x, \theta' \rangle .$$

5:     **if** Communication round is initiated **then**
6:         [Agent] Report $\theta_{a,t}^\varepsilon = \theta_{a,t}$ true parameter estimate to the principal.
7:         Update statistics $V_a^{col}$ and $B_a^{col}$ as provided by the principal.
8:     **end if**
9: **end for**

---

We now set up notation (see Table 1 for a summary of notations) and describe the protocol/algorithm in more detail.

At any time $t$, we use $\tau$ to denote the time-step at which the last communication event occurred. We represent separately the statistics from self-play and from communication as follows: $V_{a,t}^{self} = \sum_{s=1}^{t} x_{a,s} x_{a,s}^\top$ is the design matrix of agent $a$ from samples from self-play only. $V_{a,t}^{col}$ is the design matrix obtained from the principal during communication (line 7). The quantities $B_{a,t}^{self}$ and $B_{a,t}^{col}$ are analogously defined. The total statistics (from both self-play and communication) that is available to agent $a$ at time $t$ is the total design $V_{a,t} = V_{a,t}^{self} + V_{a,\tau}^{col}$, and total results $B_{a,t} = B_{a,t}^{self} + B_{a,\tau}^{col}$. The agent shall use the total statistics as of the last communication event time $V_{a,\tau}, B_{a,\tau}$ to choose and play actions for the entire

round until the next communication event. Specifically, the agent computes the parameter estimate

$$\theta_{a,\tau} = (\lambda I + V_{a,\tau})^{-1} (B_{a,\tau}). \tag{2}$$

Then, the agent constructs the confidence set

$$C_{a,\tau} = \left\{ \theta' \in \mathbb{R}^d : \|\theta' - \theta_{a,\tau}\|_{\lambda I + V_{a,\tau}} \leq \beta_t(\delta) \right\}, \tag{3}$$

where $\beta_t(\delta)$ is the radius for a confidence parameter $\delta$ (see Appendix Eq. 7). The agent uses this confidence set $C_{g,\tau}$ to choose the arm to play at all time-steps before the next communication event (line 4). Since, we do not update the confidence set every single time-step, the algorithm is said to follow a 'low-switching' or 'deferred update' paradigm where feedback from playing arms is processed in batches only during communication events.

A strategic agent might, however, not follow algorithm to play arms and instead devise their own strategy to 'free-ride' (defn. 1) on the information provided by the principal. Let $V_{a,\tau}$ be the design matrix that agent $a$ would have at time $\tau$ if he completely adheres to the prescribed protocol LsHON. Let $V_{a,\tau}^{\varepsilon} = V_{a,\tau} + \varepsilon_{a,\tau}^{V}$ be the design matrix as a result of agent $a$ deviating from this protocol, where $\varepsilon_{a,\tau}^{V}$ quantifies this deviation.

**Definition 1** (Free-rider). *A rational agent $a$ is said to* free-ride *if he explores lesser than* LsHON *requires him to for some communication event $\tau$, i.e., $V_{a,\tau}^{\varepsilon} \not\succeq V_{a,\tau}$. Equivalently, an agent with any strategy $\varepsilon_{a,\tau}^{V} \not\succeq 0$ that is not positive semi-definite is a free-rider.*

It is also possible that a strategic agent misreports their true parameter estimates, i.e., $\theta_{a,\tau}^{\varepsilon} = \theta_{a,\tau} + \varepsilon_{a,\tau}^{\theta}$, with corruption $\varepsilon_{a,\tau}^{\theta}$.

While $\varepsilon_{a,\tau}^{\theta} = 0, \varepsilon_{a,\tau}^{V} = 0$ for an honest agent, a rational agent picks $\varepsilon_{a,\tau}^{\theta}$ and $\varepsilon_{a,\tau}^{V}$ strategically.

### D.2. Principal Protocol

As per MONPEN, the principal initiates communication events as and when sufficient new statistics have been generated (line 4), i.e., when the determinant of the design matrix will have grown by a certain factor (parameterized by $c_1$) since the last communication event. During a communication event, the principal receives parameter estimates from all agents (line 6). As the parameter estimate $\theta^{\varepsilon}$ are from statistics from both self-play of agent $a$ and communication of other agents' statistics (eqn. 2), the principal recovers the agent $a$'s self-play statistics as follows:

$$B_{a,\tau}^{self} = \left( \lambda I + V_{a,\tau}^{self} + V_{a,\tau}^{col} \right) \theta_{a,\tau}^{\varepsilon} - B_{a,\tau}^{col},$$

where the principal is aware of $V_{a,\tau}^{col}, B_{a,\tau}^{col}$ as the statistics it sent $a$ in the previous communication round, and the principal can compute $V_{a,\tau}^{self}$ due to the determinism of the low-switching arm play algorithm in agent's protocol. The principal, then, shall return to every agent $a$ the complete aggregate statistics $V_{a,\tau}^{col} = \sum_{b \in \mathcal{A} \setminus \{a\}} V_{b,\tau}^{self}, B_{a,\tau}^{col} = \sum_{b \in \mathcal{A} \setminus \{a\}} B_{b,\tau}^{self}$ of all other agents, as if all agents were honest/truthful (line 8).

Next, the principal performs a statistical test to evaluate probabilistically the extent of an agent's dishonesty and assigns (line 11) a commensurate monetary penalty to the agent. Towards that, first, a $d$-dimensional vector $z_a$ is computed as a function of agent $a$'s parameter estimate and the aggregate reported parameter estimates of other agents, $\theta_{-a,\tau}^{\varepsilon}$ (line 9). The aggregate shall be computed as follows. Denote by $V_{-a,\tau} = \sum_{b \in \mathcal{A} \setminus \{a\}} V_{b,\tau}$ the aggregate design, write $B_{b,\tau}^{\varepsilon} = (\lambda I + V_{b,\tau}) \theta_{b,\tau}^{\varepsilon}$, denote by $B_{-a,\tau}^{\varepsilon} = \sum_{b \in \mathcal{A} \setminus \{a\}} B_{b,\tau}^{\varepsilon}$ the aggregate results, and finally, compute the aggregate estimate $\theta_{-a,\tau}^{\varepsilon} = (\lambda I + V_{-a,\tau})^{-1} B_{b,\tau}^{\varepsilon}$. Second, the components of $z_a$ are fed to a hand-crafted penalty function (eqn. 4) to get the monetary penalty. The reasoning behind the choice of statistical test/penalty function shall be elaborated in Section E.

Notice that the monetary penalty assigned to an agent, $P_{a,\tau}$, depends on not only their own strategy, but also on the strategies of other agents (in line 9) that agent $a$ can't observe. Thus, during any communication round (ending at time $\tau$), we can consider the following simultaneous move game induced by this protocol/mechanism: every agent $a \in \mathcal{A}$ picks their strategy $(\varepsilon_{a,\tau}^{\theta}, \varepsilon_{a,\tau}^{V})$, and receives a utility of negative of the penalty $P_{a,\tau}$. Write short-hand $\varepsilon_{a,\tau} = 0$ to denote the honest strategy of $\varepsilon_{a,\tau}^{\theta} = 0$ and $\varepsilon_{a,\tau}^{V} = 0$, and short-hand $\varepsilon_a \neq 0$ to denote any free-riding dishonest strategy where $\varepsilon_{a,\tau}^{\theta} \neq 0$ or $\varepsilon_{a,\tau}^{V} \not\succeq 0$. [3]

---

[3] We abuse notation and denote 0 to be the vector of all zeros, or matrix of all zeros, or a tuple or a list of tuples of a vector and matrix both of all zeros, disambiguated by the left hand side quantity it is being equated/compared to.

---

**Algorithm 3** MONPEN - Principal's protocol with monetary penalty

---

**Parameters:** Parameter $c_1 > 0$, $c_2 > 0$ **Input:** At all communication time-steps $\tau$, the estimate $\theta^{\varepsilon}_{a,\tau}$ from every agent $a \in \mathcal{A}$.

**Output:** At all communication time-steps $\tau$, the statistics sent back $V_{a,\tau}$, $B_{a,\tau}$, and penalty $P_{a,\tau}$ for every agent $a$.

1: Most recent communication time $\tau \leftarrow 0$.
2: **for** time-step $t = 1, 2, \ldots, T$ **do**
3:     Compute $V_{a,t}$ to be the design of every agent as per Protocol 2.
4:     **if** $\det(V_{a,t}) \geq (1 + c_1)\det(V_{a,\tau})$ **then**
5:         Initiate communication round, $\tau \leftarrow t$.
6:         Get parameter estimates $\theta_{a,\tau}$ from all agents $a$.
7:         **for all** agent $a \in \mathcal{A}$ **do**
8:             **[Principal]** Aggregate and send to agent $a$ all other agent statistics $V^{col}_{a,\tau}$, $B^{col}_{a,\tau}$.          ▷ Share full information
9:             Compute $z_a := \left(V^{-1}_{a,\tau} + V^{-1}_{-a,\tau}\right)^{-1/2}\left(\theta^{\varepsilon}_{a,\tau} - \theta^{\varepsilon}_{-a,\tau}\right)$.
10:             $\forall i \in [d]$, write $x = (z^i_a)^2$, compute component-wise penalty

$$P^i = \frac{\sqrt{2\pi x}}{(x+1)^{1+c_2}} \cdot \exp\left(\frac{x}{2}\right) - \frac{1}{c_2}. \tag{4}$$

11:             **[Principal]** Set agent $a$ a monetary penalty of $P_{a,\tau} = \frac{1}{d}\sum_i P^i$.          ▷ Levy penalty
12:         **end for**
13:     **end if**
14: **end for**

---

The next section presents the theoretical results on agent incentives and regret guarantees in this protocol.

## E. Results and Discussion

In this section, we state our theoretical results, give proof sketches, and discuss their impact and significance. We defer the formal proofs to the Appendix F in the interest of space.

We establish in Theorem 3 that principal's protocol MONPEN is designed in a way such that it induces an equilibrium of honesty (adherence to protocol) among all agents when prescribed to follow protocol LSHON .

**Theorem 3.** *Let the principal declare and follow* MONPEN. *Then, for any communication round culminating at time* $\tau$, *the strategy profile of all agents being honest, i.e.,* $(\varepsilon_{a,\tau} = 0)_{a \in \mathcal{A}}$, *is a Nash Equilibrium of the game induced among agents.*

*Proof Sketch.* We want to show that when all other agents are honest, agent $a$ suffers the least penalty when they are honest. Towards this, we prove (in Lemmas 5 and 6) that expected penalty of an agent is $O(1)$ if they are honest and doesn't converge (tends to $+\infty$) if they are dishonest and free-ride. This result primarily hinges on the principal's ability to distinguish between an honest and dishonest agent, probabilistically, from their reported parameter estimates $\theta_{a,\tau}$s without actually knowing the true parameter value $\theta^*$. Building on the intuition that a free-rider's paramter estimate $\theta_{a,\tau}$ will be 'poorer' than that of honest agents' as they don't perform the requisite exploration, the principal computes a statistical quantity $z_a \in \mathbb{R}^d$ (in line 9) by comparing the estimate of an agent ($\theta_a$) with the aggregate estimate of all other agents ($\theta_{-a}$). We show that $z_a$ doesn't depend on the unknown $\theta^*$, and conditioned on the honesty of all other agents, $z_a$ follows a 0-mean Gaussian distribution with all components having a marginal variance 1 if $a$ was honest, but if $a$ was dishonest and free-riding, some component's marginal variance goes strictly above 1 (as in Claim 7). Taking advantage of this distinction, the principal computes the penalty of agent $a$ as function of these components of $z_a$ (in line 10) specifically designed such that it's expected value converges to a constant for variance 1 (for an honest agent) and doesn't converge for variance more than 1 (for a dishonest agent). □

With this explicit specification of the equilibrium strategy profile, we note it is this equilibrium behaviour that is incorporated in LSHON for the agents. Thus, the above establishes that our principal protocol MONPEN and agent algorithm LSHON pairing is Bayesian-Nash Incentive Compatible. Next, we consider the trajectory where this equilibrium occurs and analyze the bandit regret for an agent from playing the stochastic linear bandit instance.

**Theorem 4.** *[Regret of Penalty-levying Protocol at Equilibrium] In the collaborative linear bandits problem, let the principal declare and follow* MONPEN *and all agents follow* LSHON *as per the Nash Equilibrium mentioned in Theorem 3.*

*Then, the regret of every agent is*

$$R_a(T) = O\left(d\sqrt{\frac{T \log(TM)}{M}}\right).$$

*Additionally, the expected monetary penalty of every agent is $O(1)$.*

*Proof Sketch.* The key idea is that all (honest) agents can explore and contribute equally to the collaboration. This holds due to the deterministic nature, across all agents, of the arm playing Protocol 2 between two communication events. As the learning only occurs in batches during the times of communication, all agents use the same confidence set $C_{a,\tau}$ for the entire period until the next communication event. Thus, the choice of arm played (in line 4) is same across all agents leading to all agents maintain an identical design matrix $V_{a,t}$ (see Lemma 9). This establishes that all agents share the burden of exploration equally. We quantify the benefit of collaboration among $M$ agents as an improvement (reduction) in individual regret by a factor of $\sqrt{M}$ for all agents, by using this fact that all agents contribute equally to the collaboration. The remainder of the proof broadly follows the line of argument in (Abbasi-Yadkori et al., 2011). First, we upper bound the instantaneous regret of an agent $r_{a,t}$ at time $t$, and then use it to bound the total regret to get the desired bound of $R_a(T)$.  □

Our result remarkably shows that we can attain the same regret bound as that of altruistic collaboration despite all agents being rational/self-interested. (Wang et al., 2019) showed that if the agents are impicitly altruistic, then $\tilde{O}(d\sqrt{MT})$ social regret is achievable. Here, it follows from Theorem 4 that the same regret upper bound (or utility lower bound) of $\tilde{O}(d\sqrt{MT})$ is attained when agents are implicitly strategic.

If each agent operates individually without the collaboration, they can achieve an $\tilde{O}(d\sqrt{T})$ regret using the LinUCB algorithm (Abbasi-Yadkori et al., 2011). However, our Theorem 4 shows an improvement of the regret bound to $\tilde{O}(d\sqrt{T/M})$ when all agents participate in the collaboration adhering to the mentioned equilibrium behaviour. From the single agent lower bound of (Dani et al., 2008), it follows that even with communication event at every time-step and complete sharing of statistics, equivalently setup as a single agent playing for $MT$ time sequentially, the per-agent regret bound is lower bounded as $\tilde{\Omega}(d\sqrt{T/M})$. This shows that our algorithm attains minimax optimal regret guarantee, and thus, the protocol pairing is Individually Rational, i.e., the agents are better off by participating in this collaboration, despite the threat of a monetary penalty, than abstaining.

While we show the existence of a Nash equilibrium of honesty and analyse it, it is not clear if it is unique. However, note that our Nash Equilibrium (discussed in Theorem 3) leads to a per-agent optimal minimax regret as mentioned above, and it directly follows that this leads to a socially optimal regret bound of $\tilde{O}\left(d\sqrt{TM}\right)$. Owing to this optimality, even if other equilibriums exist, they can not have a better/lesser minimax social regret guarantee. In fact, this shows that no hand-crafted strategy profile for the set of agents (even if they do not constitute an equilibrium) can lead to a better minimax social regret, showing that our Nash equilibrium of honesty is socially optimal.

## F. Missing Proofs from Section E

### F.1. Proof of Theorem 3

We establish some useful claims before we present the Proof of the Theorem.

**Claim 4.** *The ridge regression estimator $\theta_{a,t}$ follows the multi-variate Gaussian distribution with mean $\theta^* - \lambda V_{a,t}^{-1}\theta^*$ and covariance $V_{a,t}^{-1} - \lambda V_{a,t}^{-2}$.*

*Proof.* Write $V_{a,t} = \sum_{s=1}^{t} x_{a,s}x_{a,s}^\top =: X_{a,t}X_{a,t}^\top$ where $X_{a,t} \in \mathbb{R}^{d \times t}$ comprises of column vectors $x_{a,s}$s. Similarly, $B_{a,t} = \sum_{s=1}^{t} x_{a,s}y_{a,s} =: X_{a,t}Y_{a,t}$, where $Y_{a,t} \in \mathbb{R}^{t \times 1}$ comprises of rows of single rewards $y_{a,s}$s. Write $\eta(\Sigma)$ to be a random vector that follows the multi-variate gaussian distribution with mean 0 and covariance $\Sigma$. For the remainder of the proof, the subscripts $a, t$ are dropped to reduce clutter. Note that the only source of randomness in $\theta$ estimate is the gaussian noise in rewards $y_{a,t}$s observed. By working our way upwards from there, we shall show the required distribution of $\theta$.

From the way $\theta$ is calculated (as in eqn. 2),

$$\theta = (\lambda I + V)^{-1} B = (XX^\top + \lambda I)^{-1}.XY$$
$$= V^{-1}.X(\theta^{*\top} X + \eta(I_t)^\top)^\top$$
$$= V^{-1}.XX^\top.\theta^* + V^{-1}.X\eta(I_t)$$
$$= V^{-1}.(V^{-1} - \lambda I)\theta^* + \eta\left((V^{-1}X.I_t.(V^{-1}X)^\top\right)$$
$$= \theta^* - \lambda V^{-1}\theta^* + \eta\left(V^{-1}XX^\top V^{-1}\right)$$
$$= \theta^* - \lambda V^{-1}\theta^* + \eta\left(V^{-1}(V^{-1} - \lambda I)V^{-1}\right) = \theta^* - \lambda V^{-1}\theta^* + \eta\left(V^{-1} - \lambda V^{-2}\right).$$

This completes the proof. Note that with $\lambda = 0$, the estimate becomes unbiased with a variance of $V^{-1}$.  □

**Theorem 3.** *Let the principal declare and follow* MONPEN. *Then, for any communication round culminating at time $\tau$, the strategy profile of all agents being honest, i.e., $(\varepsilon_{a,\tau} = 0)_{a \in \mathcal{A}}$, is a Nash Equilibrium of the game induced among agents.*

*Proof of Theorem.* As we argue the Theorem for any arbitrary communication time-step $\tau$ in this proof, to reduce clutter, we drop the subscript $\tau$ from all notations. To prove the Theorem statement, it requires to be shown that when other agents are honest, i.e., $\varepsilon_b = 0$ for all $b \in \mathcal{A} \setminus \{a\}$ (denoted using short-hand $\varepsilon_{-a} = 0$ henceforth), the agent $a$'s expected utility is maximized when he is honest by playing strategy $\varepsilon_a = 0$, i.e., $\varepsilon_a^\theta = 0$ and $\varepsilon_a^V = 0$.

To show this, we first establish, in Lemma 5, that the expected penalty of agent $a$ converges to a small constant value when $V_a^\varepsilon = V_a, \theta_a^\varepsilon = \theta_a$ for all agents $a$. Then, we establish in Lemma 6, that the expected penalty for agent $a$ fails to converge (and tends to infinity) when agent $a$ unilaterally defects, i.e., when $\varepsilon_a \neq 0$.

**Lemma 5** (Penalty of Honest Agent). $\mathbb{E}[P_a | \varepsilon_a = 0, \varepsilon_{-a} = 0] = O(1)$.

*Proof.* As all the agents are honest, we have $V_a = V_a^\varepsilon, \theta_a = \theta_a^\varepsilon$ for all $a$.

From Claim 4, we have that $\theta_a \sim \mathcal{N}\left(\theta^*, V_a^{-1}\right)$ and $\theta_{-a} \sim \mathcal{N}\left(\theta, V_{-a}^{-1}\right)$. We next observe how the mean and more importantly, the covariance of the different quantities in the statistical test behave. It follows that $\theta_a - \theta_{-a} \sim \mathcal{N}\left(0, V_a^{-1} + V_{-a}^{-1}\right)$, and subsequently (as computed in line 9) $z \sim \mathcal{N}(0, I_d)$. Writing $z = (z^1, \dots, z^d)$, we observe all $z^i$s are i.i.d. standard normal random variables. Then, for all $i \in [d]$, we have $x = (z^i)^2$ follows the chi-squared distribution with 1 degree of freedom whose pdf we denote by $f_X$. Denote by $P(x)$ the penalty realised for a certain $x$ (as in line 10). For any $i \in [d]$, the expected component-wise penalty $P^i$ is given by

$$\mathbb{E}\left[P^i \middle| \left(\varepsilon_a^\theta = 0, \varepsilon_a^V = 0\right), \varepsilon_{-a} = 0\right] = \int_{-\infty}^\infty P(x) f_X(x) dx$$
$$= \int_0^\infty \left(\frac{\sqrt{2\pi x}}{(x+1)^{1+c_2}} \exp\left(\frac{x}{2}\right)\right) \times \left(\frac{1}{\sqrt{2\pi x}} \exp\left(\frac{-x}{2}\right)\right) dx - \frac{1}{c_2}$$
$$= \int_0^\infty \frac{1}{(x+1)^{1+c_2}} dx - \frac{1}{c_2} = 0 = O(1),$$

for any choice of $c_2 > 0$. Thus, the total expectation of total penalty $P_a = \frac{1}{d} \sum_i P^i$ is $O(1)$.  □

**Lemma 6** (Penalty of Dishonest Agent). $\mathbb{E}[P_a | \varepsilon_a \neq 0, \varepsilon_{-a} = 0]$ *does not converge/exist (tends to $+\infty$).*

*Proof.* We show this lemma in two steps. First, we analyze the $d$-dimensional vector $z_a$ (computed in line 9) and arrive at its covariance based on the $\varepsilon_a^\theta, \varepsilon_a^V$ values. Second, we argue based on this covariance that the component-wise penalty (computed in line 10) does not converge/tends to infinity for some component $P^i$.

As all other agents are honest ($\varepsilon_{-a} = 0$), we have $V_b = V_b^\varepsilon, \theta_b = \theta_b^\varepsilon$ for all $b \in \mathcal{A} \setminus \{a\}$. Consequently, $V_{-a} = V_{-a}^\varepsilon$, $\theta_{-a} = \theta_{-a}^\varepsilon$. However, as agent $a$ is dishonest/free-rides ($\varepsilon_a \neq 0$), we have $\varepsilon_a^\theta \neq 0$ or $\varepsilon_a^V \not\equiv 0$. Recollect, $V_a^\varepsilon = V_a + \varepsilon_a^V$ and $\theta_a^\varepsilon = \theta_a + \varepsilon_a^\theta$.

From Claim 4, it follows that $\theta_{-a} \sim \mathcal{N}\left(\theta^*, V_{-a}^{-1}\right)$ and also $\theta_a \sim \mathcal{N}\left(\theta^*, (V_a^\varepsilon)^{-1}\right)$. Consequently $\theta_{-a} \sim \mathcal{N}\left(\theta^*, V_{-a}^{-1}\right)$ and $\theta_g \sim \mathcal{N}\left(\theta + \varepsilon_a^\theta, (V_a^\varepsilon)^{-1}\right)$. We next observe how the mean and the covariance of the different quantities in the statistical test behave. It follows that $\theta_a - \theta_{-a} \sim \mathcal{N}\left(\varepsilon_a^\theta, (V_a^\varepsilon)^{-1} + V_{-a}^{-1}\right)$. Write a short-hand $\overline{c} := \left(V_a^{-1} + V_{-a}^{-1}\right)^{-1/2} \varepsilon_a^\theta$. Then, the vector $z_a$ (computed in line 9) obeys

$$z_a \sim \mathcal{N}\left(\overline{c}, \left(V_a^{-1} + V_{-a}^{-1}\right)^{-1/2} \left((V_a^\varepsilon)^{-1} + V_{-a}^{-1}\right)\left(V_a^{-1} + V_{-a}^{-1}\right)^{-1/2}\right). \tag{5}$$

Denote by $S$ the covariance matrix of $z_a$ (in eqn. 5). Writing $z_a = (z^1, \ldots, z^d)$, we show in Claim 7 that some component $z^i$ has marginal variance larger than 1.

**Claim 7.** *When $\varepsilon_g^V \not\equiv 0$, it holds that $\exists i \in [d] : S^{ii} > 1$.*

*Proof.* The covariance matrix $S$ is simplified as follows:

$$\left(V_a^{-1} + V_{-a}^{-1}\right)^{-1/2} \left((V_a^\varepsilon)^{-1} + V_{-a}^{-1}\right)\left(V_a^{-1} + V_{-a}^{-1}\right)^{-1/2}$$

$$= \left(V_a^{-1} + \frac{V_a^{-1}}{M-1}\right)^{-1/2} \left((V_a^\varepsilon)^{-1} + \frac{V_a^{-1}}{M-1}\right)\left(V_a^{-1} + \frac{V_a^{-1}}{M-1}\right)^{-1/2}$$

$$= \left(\frac{M}{M-1}\right)^{-1} V_a^{1/2} \left((V_a^\varepsilon)^{-1} + \frac{V_a^{-1}}{M}\right) V_a^{1/2}$$

$$= \frac{1}{M}I + \frac{M-1}{M} V_a^{1/2}(V_a^\varepsilon)^{-1} V_a^{1/2} \tag{6}$$

The second line follows from $V_{-a} = (M-1)V_a$ as the anticipated designs are identical across agents due to the determinism of the prescribed algorithm between two communication rounds.

The assumption $V_a^\varepsilon \not\succeq V_a$ leads to $I \not\succeq V_a(V_a^\varepsilon)^{-1}$ and then to $\det(V_a(V_a^\varepsilon)^{-1}) > 1$. Thus, some diagonal element of $V_a^{1/2}(V_a^\varepsilon)^{-1}V_a^{1/2}$ is larger than 1. Using this fact in line 6, we have that some diagonal element in $S$ is greater than 1. $\square$

Write within the scope of this proof $\mu = \overline{c}^i$ and $\sigma^2 = S^{ii}$ to be the mean and variance of $z^i$ for $i$ which satisfies Claim 7, with $\mu \neq 0 \vee \sigma^2 > 1$. Note that $x = (z^i)^2$ follows the generalized chi-squared distribution with 1 degree of freedom with probability density $f_X$ (written down in Appendix A.1 Claim 1). Denote by $P(x)$ the penalty realised for a certain $x$ as in line 10. The expected component-specific penalty $P^i$ is given by

$$\mathbb{E}\left[P^i \big| (\varepsilon_a^\theta \neq 0 \vee \varepsilon_a^V \not\equiv 0), \varepsilon_{-a} = 0\right] = \int_{-\infty}^{\infty} P(x) f_X(x) dx$$

$$= \int_0^\infty \left(\frac{\sqrt{2\pi x}}{(x+1)^2} \exp\left(\frac{x}{2}\right)\right) \cdot \left(\frac{1}{2\sigma\sqrt{2\pi x}} \exp\left(\frac{-(x+\mu)^2}{2\sigma^2}\right)\left[\exp\left(\frac{\sqrt{x}\mu}{\sigma^2}\right) + \exp\left(\frac{-\sqrt{x}\mu}{\sigma^2}\right)\right]\right) dx$$

$$= \int_0^\infty \left[\frac{1}{2\sigma(x+1)^2}\right] \cdot \left[\exp\left(\frac{x}{2}\left(1 - \frac{1}{\sigma^2} - \frac{\mu^2}{x\sigma^2}\right)\right)\right] \cdot \left[\exp\left(\frac{\sqrt{x}\mu}{\sigma^2}\right) + \exp\left(\frac{-\sqrt{x}\mu}{\sigma^2}\right)\right] dx,$$

which diverges as $\sigma > 1$ or $\mu \neq 0$. When $\sigma > 1$, the second term does not vanish for any value of $\mu$. When $\sigma = 1$, any $\mu \neq 0$ makes the third term not vanish.

Thus, the expectation of penalty $P_g = \frac{1}{d}\sum_i P^i$ does not converge. $\square$

From Lemmas 5, 6, we see that the expected utility of any agent can not be improved by unilaterally deviating from $\varepsilon_a = 0$. Thus, the strategy profile $(\varepsilon_a = 0)_{a \in \mathcal{A}}$ is a Nash Equilibrium. This completes the proof of the Theorem. $\square$

## F.2. Proof of Theorem 4

**Theorem 4.** *[Regret of Penalty-levying Protocol at Equilibrium]  In the collaborative linear bandits problem, let the principal declare and follow* MONPEN *and all agents follow* LSHON *as per the Nash Equilibrium mentioned in Theorem 3. Then, the regret of every agent is*

$$R_a(T) = O\left(d\sqrt{\frac{T\log(TM)}{M}}\right).$$

*Additionally, the expected monetary penalty of every agent is $O(1)$.*

We establish some useful lemmas that are used in the proof of the Theorem.

Recollect from Eqn. 3 the confidence set of agent $a$ at time $t$ (where $\tau$ is the time of last communication event) is given by $C_{a,\tau} = \left\{\theta' \in \mathbb{R}^d : \|\theta' - \theta_{a,\tau}\|_{\lambda I + V_{a,\tau}} \le \beta_t(\delta)\right\}$, where

$$\beta_t(\delta) = \sqrt{2\log\left(\frac{\det(V_{a,t})^{1/2}\det(\lambda I)^{1/2}}{\delta}\right)} + \sqrt{\lambda} = \tilde{O}\left(\sqrt{d\log(Mt/\delta)}\right). \tag{7}$$

**Lemma 8** ('Good event' occurs with high probability)**.** *For $\delta \in (0, 1/M)$, the event*

$$\mathcal{G} = \{\forall a \in \mathcal{A}, \forall t \in [T] : C_{a,t} \ni \theta^*\}$$

*occurs with probability $1 - \delta M$.*

*Proof.* We have in the single agent setting (by Theorem 2 of (Abbasi-Yadkori et al., 2011)) that with probability $1 - \delta$, $\theta^* \in C_{a,t}$ at all times $t$. By taking a union bound for all $M$ agents over the probability of failure (i.e., $\theta^* \notin C_{a,t}$), we arrive at the lemma statement. $\square$

Recollect $V_{a,t}^{self} := \sum_{s=1}^{t} x_{a,s}x_{a,s}^\top$ is the design of agent $a$ from self-play (without including information obtained through collaboration). Write $V_{\mathcal{A},t}^{self} := \sum_{a \in \mathcal{A}} V_{a,t}^{self}$ to be the global/social design at time $t$.

**Lemma 9** (Equality of design across agents)**.** *When the Principal (follows Prot. 3 and) shares all statistics with agents and all agents are honest (adhere t0 Prot. 2), then, for every time-step $t \in [T]$, the design matrix $V_{a,t}^{self}$ is the same across all agents $a \in \mathcal{A}$.*

*Proof.* Consider any two agents $a, b \in \mathcal{A}$. Denote $\tau_1, \tau_2, \ldots, \tau_q$ to be the time-steps of the $q$ communication events that occur in the entire time horizon. Additionally, write $\tau_0 = 0, \tau_{q+1} = T$.

We show the desired result by a sequence of two inductions, one on variable $i \in [q]$, and then on variable $t \in [T]$.

Let $H_1(i)$ denote the hypothesis that $B_{a,\tau_i} = B_{b,\tau_i}$. Initially, $B_{a,0} = 0, B_{b,0} = 0$, thus $H_1(0)$ is true. Assume $H_1(k)$ is true, we have $B_{a,\tau_k} = B_{b,\tau_k}$. All agents separately play arms in the period $[\tau_k + 1, \tau_{k+1}]$, until the next communication event at $\tau_{k+1}$. At time $\tau_{k+1}$, every agent $a$ shares the true parameter estimate $\theta_{a,\tau_{k+1}}$ as per Prot. 2 to the principal. The principal recovers the self-play result matrix $B_{a,\tau_{k+1}}^{self}$ of every agent and shares back $B_{a,\tau_{k+1}}^{col} = \sum_{b \in \mathcal{A}\setminus\{a\}} B_{b,\tau_{k+1}}^{self}$ the sum of results of all other agents. Thus, every agent $a$ has $B_{a,\tau_{k+1}} = B_{a,\tau_{k+1}}^{self} + B_{a,\tau_{k+1}}^{col} = B_{a,\tau_{k+1}}^{self} + \sum_{b \in \mathcal{A}\setminus\{a\}} B_{b,\tau_{k+1}}^{self} = \sum_{b \in \mathcal{A}} B_{b,\tau_{k+1}}^{self}$, which is the same quantity independent of specific agent $a$. Thus, $B_{a,\tau_{k+1}} = B_{b,\tau_{k+1}}$ and $H(k+1)$ is true. By induction, we have $H_1(i)$ is true (i.e., $B_{a,\tau_i} = B_{b,\tau_i}$) for all $i = [q]$. By an indentical argument, we have that $V_{a,\tau_i} = V_{b,\tau_i}$ for all $i = [q]$.

Next, let $H_2(t)$ denote the hypothesis that $V_{a,t'}^{self} = V_{b,t'}^{self}$ for all $0 \le t' \le [t]$. Note that initially $V_{a,0}^{self} = V_{b,0}^{self} = 0$. Thus, $H_2(0)$ is true. Assume $H_2(k)$ is true, we have $V_{a,0}^{self} = V_{b,0}^{self}, V_{a,1}^{self} = V_{b,1}^{self}, \ldots, V_{a,\tau}^{self} = V_{b,\tau}^{self}, \ldots, V_{a,k}^{self} = V_{b,k}^{self}$. At time $k+1$, let $\tau$ be the time of the most recent communication event. We have that $V_{a,\tau} = V_{b,\tau}$ and $B_{a,\tau} = B_{b,\tau}$ from the first part of this proof. By the agents' Prot. 2, we have that for every agent $g$, the confidence set $C_{g,\tau}$ is a function of the design $V_{g,\tau}$ and results $B_{g,\tau}$. Thus, $C_{a,\tau} = C_{b,\tau}$. Consequentially, at time $k+1$, the arm played $\left(\arg\max_{x \in \mathcal{X}_{.,k+1}, \theta \in C_{.,\tau}} \langle a, \theta \rangle\right)$ by agents $a$ and $b$ are equal/same, i.e., $x_{a,k+1} = x_{b,k+1}$. Combining this with $V_{a,k}^{self} =$

$V_{b,k}^{self}$ (by assumption for induction), this leads to $V_{a,k+1}^{self} = V_{a,k}^{self} + x_{a,k+1}x_{a,k+1}^{\top} = V_{b,k}^{self} + x_{b,k+1}x_{b,k+1}^{\top} = V_{b,k+1}^{self}$.
Thus, $H_2(k+1)$ is true. By induction, $H_2(T)$ is true, i.e., $V_{a,t}^{self} = V_{b,t}^{self}$ for all times $t \in [T]$.

As the above result holds for all pairs of agents $a, b$, this completes the proof of the Lemma. □

The corollary follows.

**Corollary 2.** *For all agents $a$, it holds that $V_{\mathcal{A},t}^{self} = M.V_{a,t}^{self}$, at all times $t$.*

We are ready to prove the Theorem now.

*Proof of Theorem.* Set $\delta = 1/T^2 M^2$. Then, $\mathcal{G}$ fails to occur with probability at most $1/MT^2$. As the cumulative/social regret is trivially upper bounded by $MT$, occurrence of this event only contributes a $O(T^{-1})$ summand to the expected cumulative/social regret, and consequently, to any single agent's expected regret. Thus, it is sufficient to analyse the regret of an agent when $\mathcal{G}$ occurs to show the required result.

The instantaneous regret at time $t$ is given by

$$
\begin{aligned}
r_{a,t} &= \langle x^*, \theta^* \rangle - \langle x_{a,t}, \theta^* \rangle \\
&\leq \max_{\theta \in C_{a,t}} \langle x^*, \theta \rangle - \langle x_{a,t}, \theta^* \rangle \\
&\leq \max_{\theta \in C_{a,t}} \langle x_{g,t}, \theta \rangle - \langle x_{a,t}, \theta^* \rangle \\
&= \langle x_{a,t}, \theta' \rangle - \langle x_{a,t}, \theta^* \rangle \\
&= \langle x_{a,t}, \theta' - \theta^* \rangle = \langle x_{a,t}, \theta' - \theta_{a,t} \rangle + \langle x_{a,t}, \theta_{a,t} - \theta^* \rangle .
\end{aligned}
$$

The second line is due to $\theta^* \in C_{g,t}$. The third line is due to $x_{g,t}$ being the optimistic (i.e., the expected reward maximizing) choice w.r.t $C_{g,t}$. We introduce $\theta' := \arg\max_{\theta \in C_{g,t}} \langle x_{g,t}, \theta' \rangle$ in the fourth line.

Next, recollect $V_{a,t}^{self} := \sum_{s=1}^{t} x_{a,s} x_{a,s}^{\top}$ to be the individual design of agent $g$ (without including information obtained through collaboration), and write $V_{\mathcal{A},t}^{self} := \sum_{a \in \mathcal{A}} V_{a,t}^{self}$ to be the global/social design at time $t$. Continuing the analysis by applying Hölder's inequality with a choice of conjugates $\lambda I + V_{\mathcal{A},t}^{self}$ and $\left( \lambda I + V_{\mathcal{A},t}^{self} \right)^{-1}$, we get

$$
\begin{aligned}
r_{a,t} &\leq \|x_{a,t}\|_{(\lambda I + V_{\mathcal{A},t}^{self})^{-1}} \|\theta' - \theta_{a,t}\|_{\lambda I + V_{\mathcal{A},t}^{self}} + \|x_{a,t}\|_{(\lambda I + V_{\mathcal{A},t}^{self})^{-1}} \|\theta_{a,t} - \theta^*\|_{\lambda I + V_{\mathcal{A},t}^{self}} \\
&= \|x_{a,t}\|_{(\lambda I + V_{\mathcal{A},t}^{self})^{-1}} \left( \|\theta' - \theta_{a,t}\|_{\lambda I + V_{\mathcal{A},t}^{self}} + \|\theta_{a,t} - \theta^*\|_{\lambda I + V_{\mathcal{A},t}^{self}} \right) \\
&\leq \|x_{a,t}\|_{(\lambda I + V_{\mathcal{A},t}^{self})^{-1}} \cdot \frac{\det(\lambda I + V_{\mathcal{A},t}^{self})}{\det(\lambda I + V_{a,\tau})} \left( \|\theta' - \theta_{a,t}\|_{\lambda I + V_{a,\tau}} + \|\theta_{a,t} - \theta^*\|_{\lambda I + V_{a,\tau}} \right) \\
&\leq \|x_{a,t}\|_{(\lambda I + V_{\mathcal{A},t}^{self})^{-1}} \cdot \frac{\det(\lambda I + V_{\mathcal{A},t}^{self})}{\det(\lambda I + V_{g,\tau})} . 2\beta_t(\delta) \\
&\leq \|x_{a,t}\|_{(\lambda I + V_{\mathcal{A},t}^{self})^{-1}} . 2(1 + c_1)\beta_t(\delta) \\
&= \sqrt{\frac{1}{M}} \|x_{a,t}\|_{(\frac{\lambda}{M} I + V_{a,t}^{self})^{-1}} . 2(1 + c_1)\beta_t(\delta) = c_3 \frac{\beta_t(\delta)}{\sqrt{M}} . \|x_{g,t}\|_{(\frac{\lambda}{M} I + V_{a,t}^{self})^{-1}} \\
&\leq c_3 \frac{\beta_t(\delta)}{\sqrt{M}} . \min \left\{ 1, \|x_{a,t}\|_{(\frac{\lambda}{M} I + V_{a,t}^{self})^{-1}} \right\} .
\end{aligned}
\tag{8}
$$

Here, the third line is due to $\lambda I + V_{\mathcal{A},t}^{self} \succeq \lambda I + V_{a,\tau}$ (used as in Lemma 3). The fourth line is due to the radius term used in construction of $C_{g,t} \ni \theta', \theta^*$. The fifth line is from two things: first, $V_{a,\tau} = V_{\mathcal{A},\tau}^{self}$ during times of communication events, second, the condition for Principal initiating a communication event (line 4). The sixth line is from $V_{\mathcal{A},t}^{self} = M.V_{a,t}^{self}$ as in Corollary 2. The seventh line is by trivially upper bounding $r_{a,t} \leq 2$.

990 Next, we aggregate the total regret of an agent $a$ as follows:

$$R_a = \sum_{t=1}^{T} r_{a,t} \leq \sqrt{T \sum_{t=1}^{T} r_{a,t}^2}$$

$$\leq c_3.\beta_T(\delta)\sqrt{\frac{T}{M}}.\sqrt{\sum_{t=1}^{T} \min\left\{1, \|x_{a,t}\|^2_{\left(\frac{\lambda}{M}I + V_{a,t}^{self}\right)^{-1}}\right\}}$$

$$\leq c_3.\beta_T(\delta)\sqrt{\frac{T}{M}}.\sqrt{2d\log\left(\frac{d\lambda/M + T}{d\lambda/M}\right)}$$

$$\leq c_3.\left(\sqrt{2\log\left(\frac{\det(V_{a,t})^{1/2}\det(\lambda I)^{1/2}}{\delta}\right)} + \lambda^{1/2}\right)\sqrt{\frac{T}{M}}.\sqrt{2d\log\left(\frac{d\lambda/M + T}{d\lambda/M}\right)}$$

$$\leq c_4.\sqrt{d\log(MT)}.\sqrt{\frac{T}{M}}.\sqrt{d\log(MT)} = O\left(d\sqrt{\frac{T\log(MT)}{M}}\right)$$

Here, the first line uses Jensen's inequality (on concavity of the square root function) or Cauchy-Schwarz inequality. The second line follows from Eqn. 8. The third line is due to Lemma 2 with $\gamma = \lambda/m$. The fourth line uses Eqn. 7. The fifth/final line uses $\det(V_{g,t}) \leq \left(\frac{MT}{d}\right)^d$, $d < t$, $\delta = (T^2M^2)^{-1}$. This completes the proof of the individual agent regret guarantee mentioned in the Theorem.

The expected monetary penalty of an individual agent is $O(1)$ as already shown in Lemma 5 when all agents are honest. This completes the proof of the Theorem. □

## G. Limitations

We discuss the limitations of our work in this section. Firstly, our work involves monetary penalty in the design of mechanism. While it is possible to execute in a real-world environment, it would be interesting to see if one can design a payment-free mechanism. Secondly, we assume that the agents have the same set of actions at each time-step. While our results allow some relaxation of this assumption, it would be more realistic to allow for arbitrarily different actions sets for each agents. With arbitrarily different action sets, the very benefit of collaboration, in fact, is not clear. Lastly, our work does not consider competition between different agents which is a possibility in a real-world scenario.

## H. Additional Experiments Details and Results

In this section, we provide additional experiment results and explanations that were not a part of the main paper.

### H.1. Synthetic experiments

In the synthetic environment, we generate true parameter $\theta^*$ and the action sets $\{\mathcal{X}_t\}_t$ randomly as follows: each component of $\theta^*$ and each component of each action in an action set $\mathcal{X}_t$ is assigned a value uniformly at random in $[-1, 1]$. We consider an embedding dimension of $d = 50$ and actions sets of size $|\mathcal{X}_t| = 400$ at all times $t$ for our synthetic experiments.

**Presence of free-rider.** The purpose of this experiment is to demonstrate the usefulness of our Protocol for multi-agent stochastic linear bandits problem in the following sense. Existing algorithms are not immune to the presence of strategic agents, i.e., a strategic agent will be able to 'free-ride' leading to a bad social outcome over time due to a collective lack of exploration. On the other hand our algorithm will be able to levy a monetary penalty to the agents, the penalty serves as a deterrent to such 'free-riding' and creates an incentive to be truthful. To elucidate this, in our collaborative learning setup, we let one of the agents 'free-ride' and analyse how the regret and penalty of the agents behave. Specifically, the free-rider agent, say $a$, plays the best arm according to the $\theta_{a,\tau}$ based on total statistics from self-play and from communication by the principal during the previous communication event, and does not perform exploration of his own. Specifically, he plays $x_{a,t} = \arg\max_{x \in \mathcal{X}_t} \langle x, \theta_{a,\tau} \rangle$, while all other agents fulfill the exploration as dictated in (line 4 of) LSHON. All agents, including the free-rider, report their current estimate $\theta_{a,t}$ honestly.

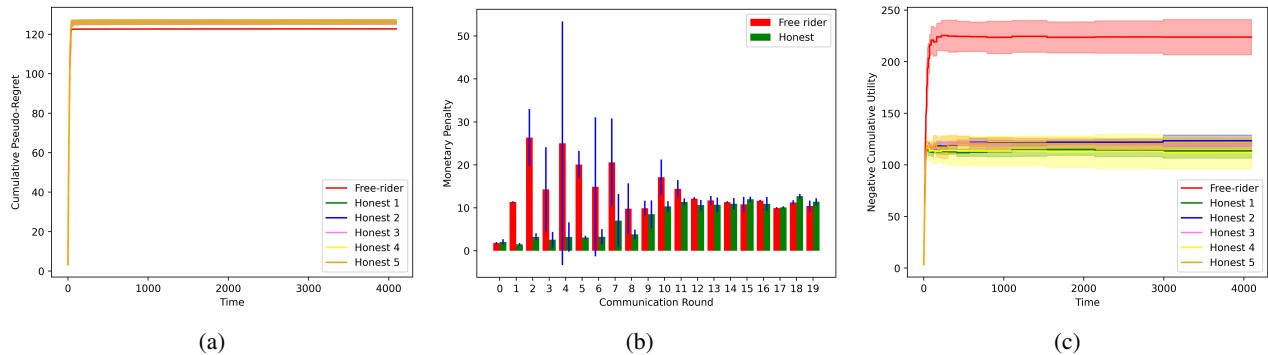

(a)                                    (b)                                    (c)

Figure 2: **Synthetic problem instance** All sub-figures (a, b, c) plot the corresponding metrics as described in Fig. 1.

The results of the synthetic experiment is plotted in Fig. 2. We plot the cumulative regret, round-wise penalty, and cumulative utility of a bunch of agents. The negative of the utility of an agent is the sum of his regret and his monetary penalty minus the average penalty of all agents. We draw identical insights from the results of the two experiments as described in Section 4.

