# OpenReview forum: "Collaborative Learning under Strategic Behavior: Mechanisms for Eliciting Feedback in Principal-Agent Bandit Games"
_ICML.cc/2024/Workshop/Agentic_Markets — Agentic Markets @ ICML'24 Poster_

### Official Review · Reviewer_uzu9 · 2024-06-10
**Review of Submission 30**

**Rating:** 7
**Confidence:** 3

**Review:**

**Quality**

Connecting results in regret minimization and collaborative MABs to avoid free-riding is an interesting problem, and the contributions of this paper show that it is possible to disincentivize free riding in collaborative bandit problems. The algorithm and protocol proposed within this paper are well argued for on both a theoretical and practical level. Moreover, the assumptions required to establish the theoretical results are based on existing ideas in the literature. Overall, I think this paper is of sound technical quality for the workshop.

**Clarity**

The paper is well written and expresses a strong motivation for both the problem tackled and why the corresponding results are important. The sketches of the protocols and experiments are nicely summarized and give a fairly clear picture of the main ideas.

**Originality**

The key results of the paper concerning the regret analysis of agents using LSHON are an extension of the result given in Abbasi-Yadkori et al., 2011, and indeed the proof technique does not seem novel. However, the principal's protocol MONPEN is designed in such a way as to induce a non-free-riding equilibrium if agents all follow LSHON, and the arguments in the proof of Thm 3 are to my knowledge original for the framework.

**Significance**

Overall, the experiments and results are fairly convincing in showing that in certain cases, collaboration can be the equilibrium behavior of strategic agents. However, this comes with the caveat that a) the agents all need to have the same action sets and must follow the LSHON protocol, and b) the MONPEN protocol needs to be usable (i.e. the system has to be amenable to monetary penalties). As an initial foray into understanding the interplay between collaboration and mechanism design in MAB, I believe this is a good paper worthy of acceptance to the workshop. In future work, it would be nice to see if the above restrictions can be relaxed. For instance, can the agents have complementary actions within different actions sets that still amount to collaboration? Can MONPEN be generalized to settings where the penalty is not strictly monetary in nature?

---

### Official Review · Reviewer_aVYD · 2024-06-12
**Interesting initial results on mechanisms for disincentivizing free-riding in collaborative multi-agent bandit setting.**

**Rating:** 6
**Confidence:** 3

**Review:**

**Summary**:

This paper studies a collaborative, multi-agent linear bandit problem with a central mediator. The main contribution is on designing mechanisms to disincentivize strategic/dishonest agents from “free-riding” on the information received from other agents, without themselves performing exploration.

The main contribution is a pair of mechanisms (MonPen for the mediator/principal and LsHon for the agents) with the following guarantees:

(i) the strategy profile of all agents simultaneously following the algorithm LsHon is a Nash EQ (when the principal follows MonPen).

(ii) when all agents follow LsHon, they each obtain a sublinear regret of roughly d \sqrt{T/M} (where d is the dimension of the action space, and M is the number of agents).

The MonPen mechanism allows the principal to levy monetary penalties on suspected free-riding agents, and the LsHon algorithm is roughly based on the low-switching variant of LinUCB of Abbasi-Yadkori+2011.

The authors also show experimentally that the proposed mechanisms incentivize free-riding agents to behave more honestly over time.

**Pros:**

+ the question of provably disincentivizing “freeriding” in collaborative multi-agent bandit settings seems interesting and under-explored.
+ the simultaneous Nash and sublinear regret bound guarantees are compelling.

**Cons/questions:**
- The experimental results are somewhat limited: it would be interesting to see more results when the number of free-riding agents is more than 1.
- It could be helpful to have a more detailed/complete proof of Theorem 4 (regret guarantee) in the appendix, to more easily understand where the dependence on M arises. Also, perhaps the authors could mention the regret guarantees for honest agents (following LsHon) when there are still \alpha fraction of dishonest agents? How does the regret guarantee scale with \alpha and M in that case?